# Graph-Guided Scene Reconstruction from Images with 3D Gaussian Splatting

**Chong Cheng**[1]* **Gaochao Song**[2]* **Yiyang Yao**[3] **Qinzheng Zhou**[4]
**Gangjian Zhang**[1] **Hao Wang**[1]†
[1]HKUST(GZ) [2]HKU [3]SCUT [4]UC Berkeley

## Abstract

This paper investigates an open research challenge of reconstructing high-quality, large 3D open scenes from images. It is observed existing methods have various limitations, such as requiring precise camera poses for input and dense viewpoints for supervision. To perform effective and efficient 3D scene reconstruction, we propose a novel graph-guided 3D scene reconstruction framework, GraphGS. Specifically, given a set of images captured by RGB cameras on a scene, we first design a spatial prior-based scene structure estimation method. This is then used to create a camera graph that includes information about the camera topology. Further, we propose to apply the graph-guided multi-view consistency constraint and adaptive sampling strategy to the 3D Gaussian Splatting optimization process. This greatly alleviates the issue of Gaussian points overfitting to specific sparse viewpoints and expedites the 3D reconstruction process. We demonstrate GraphGS achieves high-fidelity 3D reconstruction from images, which presents state-of-the-art performance through quantitative and qualitative evaluation across multiple datasets. Project Page: `https://3dagentworld.github.io/graphgs/`.

## 1 Introduction

3D scene reconstruction aims to transform 2D images into realistic 3D scenes. This technology has many practical applications such as Augmented Reality (AR) and Virtual Reality (VR) (Cheng et al., 2023; Guo et al., 2023; Mi & Xu, 2023; Xiangli et al., 2022; Xu et al., 2023). The emergence of Neural Radiance Fields (NeRF) (Mildenhall et al., 2021) and 3D Gaussian Splatting (3DGS) (Kerbl et al., 2023) enables differentiable novel-view synthesis and real-time rendering (Luiten et al., 2024; Wu et al., 2023; Yang et al., 2023c). Recent studies have applied NeRF and 3DGS to unbounded environments such as street views and urban areas (Turki et al., 2022; Tancik et al., 2022; Wang et al., 2023c; Lin et al., 2024; Yan et al., 2023), broadening the application scenario.

However, achieving high-quality 3D scene reconstruction remains a challenging task. It is observed existing works require precise camera poses for input and dense viewpoints for supervision (Lin et al., 2024; Chen et al., 2024; Guo et al., 2023). Notably, public benchmark datasets such as those referenced in (Geiger et al., 2012; Turki et al., 2022; Sun et al., 2020) often experience pose inaccuracies and sparse viewpoints for boundaries. These issues are typically attributed to motion-induced vibrations and limitations in the positioning equipment used in vehicles or drones. Moreover, if one would like to use mobile devices such as smartphones and DSLR cameras to capture images for 3D reconstruction, obtaining accurate camera poses and dense viewpoints can be challenging and troublesome.

Although tools like COLMAP have been developed to lower the entry barrier for 3D reconstruction by providing pose estimation, the heavy optimization costs and camera matching failures limit their practicality (Schönberger & Frahm, 2016; Schönberger et al., 2016b). Additionally, methods that incorporate pose estimation strategies for object reconstruction, struggle with broader scene applications (Yang et al., 2023b; Jain et al., 2021; Wang et al., 2023a; et al., 2023a). The limited number of viewpoints and the vast number of images in outdoor scenes further complicate the issue.

---

*Equal contribution.
†Corresponding author.

To address the challenges of reconstructing large open scenes from uncalibrated images, this paper proposes GraphGS, a framework specifically tailored for large 3D open scene reconstruction. GraphGS utilizes 3DGS combined with our proposed spatial prior-based scene structure estimation to enhance the speed and accuracy of pose estimation, even for image collections comprising thousands of images. During the process of scene structure estimation, the matching relationship of cameras is also recorded, which is represented as a camera graph. The camera graph provides useful topology information for scene cameras and benefits for gaussian optimization.

GraphGS innovatively proposes to apply the constructed camera graph to guide the optimization process of large 3DGS scenes, through multi-view consistency constraints and adaptive sampling strategy. This ensures a more accurate 3D Gaussian point distribution. Besides, our proposed approach prevents Gaussian points from overfitting to given sparse viewpoints, thereby enhancing reconstruction quality. Furthermore, our adaptive sampling strategy reduces the number of iterations for camera graph nodes, significantly accelerating the reconstruction of large open 3DGS scenes and reducing training time by nearly 50%.

The main contributions of this paper are summarized as follows:

- We introduce 3DGS in conjunction with spatial prior-based structure estimation method to efficiently and accurately estimate structures from uncalibrated images.
- Through our proposed camera graph-guided 3D Gaussian optimization, GraphGS not only improves the reconstruction quality, but also greatly accelerates the reconstruction process.
- With GraphGS framework, our method achieves state-of-the-art (SOTA) performance in several large benchmarks without using ground truth camera poses.

## 2 RELATED WORK

### 2.1 SCENE RECONSTRUCTION

The advent of NeRF (Mildenhall et al., 2021) has ushered in a golden age for 3D scene construction. Numerous studies have improved its efficiency (Hedman et al., 2021; Müller et al., 2022; Reiser et al., 2023) and generalization (Yu et al., 2021b; Wang et al., 2021a; Chen et al., 2021). Mip-NeRF (Barron et al., 2021) and Zip-NeRF (Barron et al., 2023) have tackled aliasing issues, while InstantNGP (Müller et al., 2022) integrates grid pyramid technologies to optimize sub-volumes. UC-NeRF (Cheng et al., 2023) targets outdoor scenes, enhancing image consistency through color correction and pose refinement. StreetSurf (Guo et al., 2023) and EmerNeRF (Yang et al., 2023a) introduce novel approaches for multi-view reconstructions through disentanglement and self-guided learning. Concurrently, PVG (Chen et al., 2024) utilizes 3DGS (Kerbl et al., 2023) to advance scene reconstruction with techniques like time-dependent transparency and scene-flow smoothing.

Furthermore, to extend reconstruction techniques to larger-scale scenes, methods like Block-NeRF (Tancik et al., 2022), Mega-NeRF (Turki et al., 2022), and Switch-NeRF (Mi & Xu, 2023) employed a divide-and-conquer strategy. Mega-NeRF clustered pixels based on 3D sampling distances, Block-NeRF organized images into street blocks, and Switch-NeRF utilized a sparse network for large scene synthesis. These methods improved scalability and flexibility but faced limitations in real-time rendering of large outdoor environments. VastGaussian (Lin et al., 2024) incorporated 3DGS to enhance detail presentation and rendering speed in large scenes. While these methods have advanced scene reconstruction, they typically rely on precise camera poses and initial data like LiDAR, which can be difficult to obtain in real-world applications, especially in expansive outdoor settings.

### 2.2 POSE OPTIMIZATION

To address issues with pose accuracy, many studies seek to bypass the slow and occasionally imprecise COLMAP process by concurrently optimizing camera pose and scene representation using the original MLP-based NeRF (Wang et al., 2021b), such as GARF (Chng et al., 2022) and BARF (Lin et al., 2021). Joint-TensoRF(Cheng et al., 2024) focuses on refining camera poses and 3D scenes using decomposed low-rank tensors. These methods have been proven effective in recovering object structures and poses from imperfect or unknown camera positions, although their application

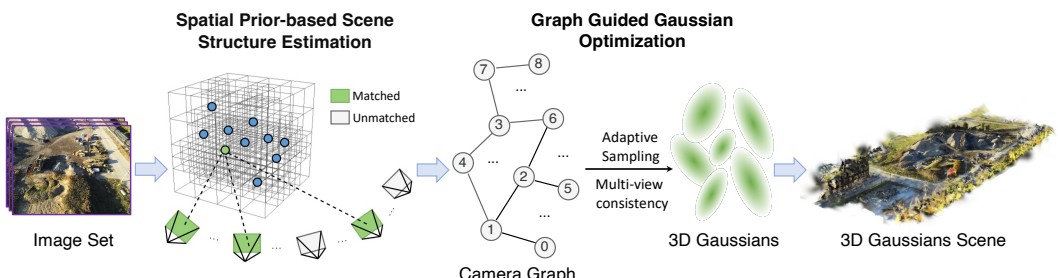

Figure 1: Framework of the GraphGS method for efficient large 3D scene reconstruction. The process begins with spatial prior-based structure estimation, followed by octree-based efficient organization of initialization points. The camera graph is obtained at the end of structure estimation, which contains topology information of scene camera. The information in camera graph will be further used for the following gaussian optimization.

to broader scene reconstruction remains challenging. In 3DGS, COLMAP (Schönberger & Frahm, 2016; Schönberger et al., 2016b) is utilized for pose reconstruction and generating sparse initial points via Structure-from-Motion (SfM) (Schönberger & Frahm, 2016). However, COLMAP's dense matching time increases exponentially with the number of images, and its success rate is limited, which poses significant challenges for outdoor scene reconstruction.

Building on the discussions above, this paper is dedicated to proposing a low-pose-requirement, rapid, high-precision 3D reconstruction method suitable for both general and large scenes.

## 3 METHOD

As shown in Fig. 1, we firstly design spatial prior-based scene structure estimation to improve efficiency of SfM pipeline with the input of thousands of images. During the process, we record the matching information of cameras and form a camera graph. The graph contains topology information of scene cameras, which is suitable for guiding gaussian optimization to improve reconstruction quality and efficiency. In addition, to accelerate the training process and decreases GPU memory consumption, an octree point initialization strategy is introduced to reduce initialization points.

### 3.1 SPATIAL PRIOR-BASED STRUCTURE ESTIMATION

In the reconstruction of 3DGS scenes from thousands to tens of thousands of images, rapidly and accurately obtaining camera poses represents a significant challenge. Traditional Structure from Motion (SfM) pipeline(COLMAP (Schönberger & Frahm, 2016; Schönberger et al., 2016b)), typically require days to accurately process such extensive datasets. We locate this problem in exhaustive matcher of COLMAP pipeline, which takes all possible image pairs for feature matching and causes time bottleneck. To address this issue, we have developed an innovative spatial analysis framework for estimating scene structure. Initially, we acquire approximate camera poses from pre-trained fast relative pose estimation models. Subsequently, utilizing concentric nearest neighbor pairing (Sec. 3.1.1) and quadrant filter (Sec. 3.1.2), we filter and prioritize image pairs that are crucial for accurate camera pose estimation. This approach dramatically reduces computational load, facilitating swift and precise reconstructions that transform processes from days-long endeavors to tasks completed within hours. Additionally, we have constructed a camera graph to guide the optimization process in 3DGS, further enhancing the efficiency and precision of the reconstruction.

### 3.1.1 CONCENTRIC NEAREST NEIGHBOR PAIRING

Compared to selecting all possible image pairs, opting for a sparser set can significantly accelerate the process of scene structure estimation. However, improper selections may lead to failures in matching cameras with the scene during the estimation phase. Our principle is, the distribution of selected image pairs should reflect both local and global structure. To handle this issue, we propose concentric nearest neighbor pairing. For camera $c_i, c_j$ from $N$ cameras altogether, we define $\mathbf{s}(i, j)$

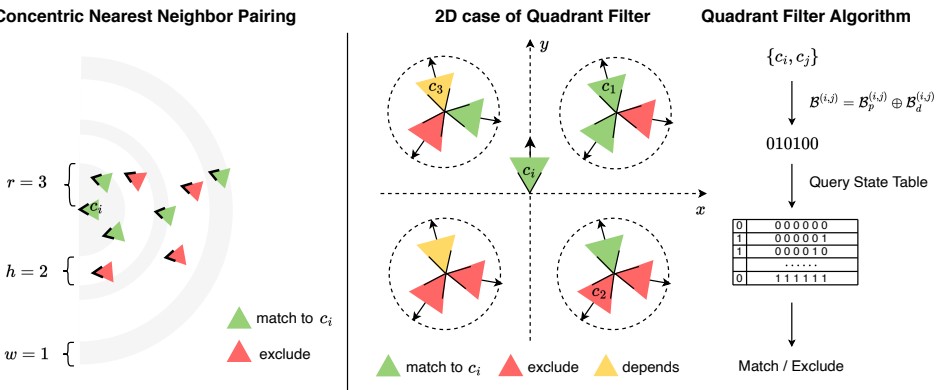

Figure 2: Illustration of spatial prior-based structure estimation, including two parts: Concentric Nearest Neighbor Pairing (left) and Quadrant Filter (right). For Concentric Nearest Neighbor Pairing, we first select $c_i$'s nearest $r$ cameras for matching to guarantee stability of local bundle adjustment, then we select $w$ cameras from every $h + w$ camera based on distance order, forming a series of concentric circles. For quadrant filter of 2D case, camera $c_i$ is posed at the center of the coordinate system, pointing towards the y-axis. For other cameras (we show 12 cameras), the relative position and orientation to $c_i$ contains $4 \times 4 = 16$ states.

as the sorted order from nearest to farthest of camera $c_j$ relative to camera $c_i$, then the set of all matched pairs $S_{all}$ is given by:

$$S_{all} = \bigcup_{i=1}^{N} (S_{\text{neighbor}}^{(i)} \cup S_{\text{concentric}}^{(i)} \cup S_{\text{connection}}^{(i)}) \tag{1}$$

$$S_{\text{neighbor}}^{(i)} = \{(c_i, c_j) | s(i,j) \le r, j = 1, 2, ..., N, j \ne i\}$$

$$S_{\text{concentric}}^{(i)} = \{(c_i, c_j) | 0 \le (s(i,j) - r)\%(h+w) < w, j = 1, 2, ..., N, j \ne i\}$$

$$S_{\text{connection}}^{(i)} = \{(c_i, c_j) | j = i - 1, j \ge 1\}$$

where the process of $S_{\text{neighbor}}^{(i)}$, $S_{\text{concentric}}^{(i)}$ are illustrated in Fig. 2 (left). Specifically, we traverse all of the cameras from 1 to $N$ (supposing the current camera is $c_i$) and calculate which camera $c_j$ (from another loop) should form a matching pair with $c_i$. Then we have the following steps: (1) For $c_i$, we sort other cameras based on their distance to $c_i$, and add $c_i$'s nearest $r$ cameras as $S_{\text{neighbor}}^{(i)}$. (2) We add $w$ cameras from every $h + w$ cameras based on the same sorted order as $S_{\text{concentric}}^{(i)}$. (3) We add $c_{i-1}$ in the main loop to match $c_i$ as $S_{\text{connection}}^{(i)}$.

This approach allows for a smooth transition from local to global matching pairs, ensuring sparsity that enhances the efficiency of feature matching. Notice there are repetitive elements (e.g. camera pair $(c_1, c_3)$ is equivalent to $(c_3, c_1)$), we eliminate them via HashSet at the end. Also, we implement this method via K-DTree ( (Bentley, 1975)) for highly efficient storage and retrieval of camera distance relationships to save pair selection time.

**Remark.** When treating cameras as a graph node, and adding edges for every matched pair in $S_{all}$, an undirected graph will be formed, which we refer to as *camera graph*. To guarantee the success of scene structure estimation, camera graph should be *connected graph*. However, this may not be satisfied when we only have $S_{\text{neighbor}}^{(i)}$, $S_{\text{concentric}}^{(i)}$, and the proof is presented in Appendix A. To address this issue, we add additional $S_{\text{connection}}^{(i)}$ and pairs in this set will not be affected by the following quadrant filter strategy.

### 3.1.2 QUADRANT FILTER

In reconstructing 3DGS scenes from a large number of images, selecting appropriate image pairs to accelerate the estimation of scene structure is essential. Given two camera pairs which have little or

no view intersection with each other, it is inefficient to mark them as matching pairs. Besides, these pairs introduce noise for global refinement. In this section we introduce quadrant filter strategy to filter these noise pairs.

For one coarse camera $c_i$, we denote its global position as $p^{(i)} = [x_i, y_i, z_i]$ and orientation as $d^{(i)} = [v_x^{(i)}, v_y^{(i)}, v_z^{(i)}]$. Broadly speaking, for arbitrary two camera $c_i, c_j$ with varying position and orientation in 3D space, their relative position and relative orientation have 8 states respectively, totally with $8 \times 8 = 64$ states. This already provides enough information to filter noise pairs.

To record the relative position of camera $c_j$ to $c_i$, we apply binary encoding as follows:

$$\mathcal{B}_p^{(i,j)} = \{sgn(x_j - x_i), sgn(y_j - y_i), sgn(z_j - z_i)\} \tag{2}$$

with the sign function $sgn(x)$:

$$sgn(x) = \begin{cases} 1, & \text{if } x > 0 \\ 0, & \text{if } x \leq 0 \end{cases} \tag{3}$$

where $\mathcal{B}_p^{(i,j)}$ is a three-digit binary number, representing 8 quadrants. For example, when $\mathcal{B}_p^{(i,j)} = \{0,1,1\}$, it represents binary number $011$, corresponding to the 3rd quadrant. $(i,j)$ represents the position of $c_j$ relative to $c_i$, meaning $c_i$ is posed on the origin of the 3D coordinate system, pointing towards z-axis.

To record the relative orientation, a direct way is to calculate a rotation matrix to transform them in the same standard, and follow the same approach like recording relative position. However, this would be time-consuming since we have to solve a system of linear equations for every potential camera pairs. To address this issue, we have proposition as follows:

**Proposition 1** *Given two spacial vectors $d^{(i)} = [v_x^{(i)}, v_y^{(i)}, v_z^{(i)}], d^{(j)} = [v_x^{(j)}, v_y^{(j)}, v_z^{(j)}]$, their relative orientation have 8 states respectively, corresponding to 8 quadrants in 3D coordinate system. The quadrant of relative orientation can be directly calculated via one cross product and one inner product:*

$$\mathcal{B}_d^{(i,j)} = \{sgn(e_x[d_\times^{(i)}]d^{T(j)}), sgn(e_y[d_\times^{(i)}]d^{T(j)}), sgn(d^{(i)}d^{T(j)})\} \tag{4}$$

*where $\mathcal{B}_d^{(i,j)}$ is a three-digit binary number, representing 8 quadrants. $[d_\times^{(i)}]$ represents the antisymmetric matrix of cross product:*

$$[d_\times^{(i)}] = \begin{bmatrix} 0 & -v_z^{(i)} & v_y^{(i)} \\ v_z^{(i)} & 0 & -v_x^{(i)} \\ -v_y^{(i)} & v_x^{(i)} & 0 \end{bmatrix} \tag{5}$$

*and $e_x = [1, 0, 0], e_y = [0, 1, 0]$ represent unit vector of $x, y$ axis.*

*The full proof is provided in Appendix C.*

Proposition 1 provides an efficient way to record 8 states of relative orientation. After encoding both position and orientation, we concatenate them to get final pose binary encoding with 6-bit:

$$\mathcal{B}^{(i,j)} = \mathcal{B}_p^{(i,j)} \oplus \mathcal{B}_d^{(i,j)} \tag{6}$$

this enables us to directly query a predefined state table to filter the noise camera pairs. As shown in Fig. 2 (right), a 2D example, in this case, the relative position and relative orientation have 4 states respectively (that is, 4 quadrants). For different camera $c_j$ relative to $c_i$, the $c_1$ (green) should be included in matching if it appears in $S^{(i)}$ of Equation 1, while the $c_2$ (red) should be excluded. For $c_3$ (yellow), it has slight view intersection with $c_i$, for this state, we set strict/loose mode of the state table to handle it in practice. Notice even in the loose mode (we include all the yellow cameras like $c_3$ to matching), our method can also filter more than $38\%$ noise camera pairs for cameras with random poses. The detailed quadrant division of 3D coordinate system, full state table and analysis of strict/loose mode are all presented in Appendix B.

### 3.1.3 Octree Point Initialization

The design of this module is based on the observation that, in the scene reconstruction process using 3DGS, not all initial points contribute equally to the ultimate quality of the reconstruction. By efficiently managing these points, computational efficiency is optimized without compromising reconstruction quality. Drawing inspiration from PlenOctrees (Yu et al., 2021a), we employ an Octree (Meagher, 1980) structure for spatial detail management. Nodes in the octree are evaluated based on the Level of Detail (LOD) of the initial points, with those falling below a detail threshold $\tau$—deemed minimally contributory to quality—being pruned to simplify the model.

### 3.2 Graph-guided Gaussian Optimization

#### 3.2.1 Camera Graph Definition

Based on the observation that viewpoint coverage in open scenes is relatively sparse compared to indoor scenes or object reconstruction, supervision from a single viewpoint can lead to Gaussian points overfitting near specific cameras. This results in an inaccurate Gaussian distribution and learning predominantly from a single image rather than the entire scene. To address these challenges and improve the efficiency, accuracy, and robustness of 3DGS reconstruction in open scenes, we have specifically designed a weighted undirected camera graph based on structured estimation results. The construction of this graph adheres to the following rules: 1) Each node corresponds to a view camera's rotation matrix R and translation matrix T; 2) Edges are formed based on the camera pairs selected in Section 3.1. We aim to use this meticulously designed graph structure to precisely reflect the spatial relationships between cameras, thereby effectively guiding the optimization process of 3DGS in open scenes.

#### 3.2.2 Graph-guided Multi-view Consistency Constraint

We know that adjacent multi-view scenes with minimal directional differences should exhibit nearly identical photometric values (Kloukiniotis et al., 2022). Consequently, we designed edge weight $w_e(i,j)$ of camera graph to measure this directional differences as follows:

$$w_e(i,j) = \frac{e^{-k\|p^{(i)}-p^{(j)}\|_2}}{1-e^{-d^{(i)}d^{T(j)}}} \tag{7}$$

where $p^{(i)}$, $p^{(j)}$ and $d^{(i)}$, $d^{(j)}$ are refined camera position and orientation from rotation and translation matrix $R$, $T$. Take Fig. 3 for example, supposing there are 3 edges $(3,4),(3,6),(3,7)$ connecting to camera node 3. When calculating the edge weights, Eq. 7 will take relative distance $d_{36}, d_{37}, d_{34}$ and relative orientation $\theta_{36}, \theta_{37}, \theta_{34}$ into consideration, while the edge with smaller distance and smaller angle obtains larger weight.

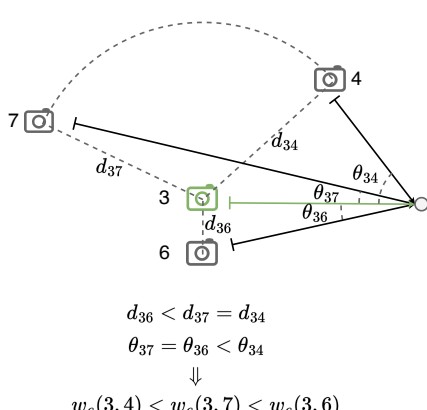

$$d_{36} < d_{37} = d_{34}$$
$$\theta_{37} = \theta_{36} < \theta_{34}$$
$$\Downarrow$$
$$w_e(3,4) < w_e(3,7) < w_e(3,6)$$

Figure 3: Example of edge weights.

Based on the edge weights, we designed a multi-view consistency photometric loss $\mathcal{L}_{\text{cons}}$ to enhance the robustness of 3DGS optimization process as follows:

$$\mathcal{L}_{\text{cons}} = \lambda \sum_{i,j} \sum_{p} \|I_i(p) - I_j(\mathbf{K}_j(R_{ji}\mathbf{K}_i^{-1}p + T_{ji}))\| \tag{8}$$

where $I_i(p)$ represents the intensity or color at pixel $p$ in the image from camera $i$. The matrices $R_{ji}$ and $T_{ji}$ describe the rotation and translation from camera $i$ to camera $j$, respectively, while $\mathbf{K}_i$ and $\mathbf{K}_j$ are the intrinsic matrices, and $\lambda$ is an adjustment factor. To obtain camera $j$ given camera $i$, we traverse all edges of camera $i$ and select the camera with the maximum edge weight to camera $i$ as target camera $j$.

Counterintuitively, while target camera $j$ may provide rendered results that are significantly worse than the ground truth images for supervision, they enhance the reconstruction outcome. We believe this may act similar to data augmentation method, helping to prevent overfitting.

| Method | Waymo | | | | KITTI | | | |
|---|---|---|---|---|---|---|---|---|
| | FPS ↑ | PSNR ↑ | SSIM ↑ | LPIPS ↓ | FPS ↑ | PSNR ↑ | SSIM ↑ | LPIPS ↓ |
| Mip-NeRF 360 (Barron et al., 2022) | 0.042 | 22.42 | 0.698 | 0.471 | 0.053 | 20.68 | 0.650 | 0.480 |
| S-NeRF (Xie et al., 2023) | 0.001 | 19.22 | 0.515 | 0.400 | 0.008 | 18.71 | 0.606 | 0.352 |
| StreetSurf (Guo et al., 2023) | 0.097 | 23.78 | 0.822 | 0.401 | 0.037 | 22.48 | 0.763 | 0.304 |
| Zip-NeRF (Barron et al., 2023) | 0.500 | 26.21 | 0.815 | 0.389 | 0.610 | 21.41 | 0.665 | 0.470 |
| UC-NeRF (Cheng et al., 2023) | 0.032 | 26.72 | 0.800 | 0.375 | 0.051 | 24.05 | 0.721 | 0.400 |
| BARF (Lin et al., 2021) | 0.041 | 9.07 | 0.235 | 1.021 | 0.071 | 10.68 | 0.250 | 0.990 |
| SPARF (et al., 2023b) | - | - | - | - | - | - | - | - |
| UP-NeRF (et al., 2023a) | 0.120 | 26.16 | 0.876 | 0.375 | - | - | - | - |
| EmerNeRF (Yang et al., 2023a) | 0.043 | 25.92 | 0.763 | 0.384 | 0.28 | 25.24 | 0.801 | 0.237 |
| 3DGS (Kerbl et al., 2023) | **63** | 25.08 | 0.822 | 0.319 | **125** | 19.54 | 0.776 | 0.224 |
| PVG (Chen et al., 2024) | 50 | 28.11 | 0.849 | 0.279 | 59 | 26.63 | 0.885 | **0.127** |
| Ours | 52 | **29.43** | **0.899** | **0.217** | 79 | **26.98** | **0.887** | 0.157 |

Table 1: Quantitative comparison of novel view synthesis results on the Waymo and KITTI. Our method demonstrates a competitive edge in rendering quality with higher FPS, PSNR, SSIM, and lower LPIPS scores, affirming its efficacy in synthesizing realistic views in comparison to the current state-of-the-art methods.

### 3.2.3 ADAPTIVE SAMPLING OPTIMIZATION

We propose an adaptive sampling strategy that dynamically adjusts sampling rates during optimization based on node importance in the graph. For nodes with less overlap with other viewpoints, sampling frequency is reduced according to their weights. To assess the importance of each node within the graph, we devise specific importance weights. We find that performing fewer iterations on certain peripheral nodes does not significantly degrade the overall quality of reconstruction and helps prevent overfitting due to insufficient supervision signals. The design of node weights primarily considers two criteria: 1) Degree centrality, which calculates the number of neighbors connected to each camera node; 2) Betweenness centrality, assessing the importance of each node across all shortest paths, where nodes with higher centrality are considered more significant. Degree centrality focuses on the activity level of nodes, while betweenness centrality is concerned with the control a node exerts or its role as a bridge. In our experiments, we found that using only betweenness centrality yielded better results, so we design node weight $w_n(i)$ based on Betweenness centrality:

$$w_n(i) = \sum_{i \neq j \neq k} \frac{\sigma_{jk}(i)}{\sigma_{jk}} \tag{9}$$

where $\sigma_{jk}$ represents the number of shortest path between node $j,k$ of camera graph. $\sigma_{jk}(i)$ represents the number of shortest path passing node $i$ between node $j,k$. After getting node weights, we set a probability function $P(i)$ for camera $c_i$ during iteration:

$$P(i) = w_n(i)/\text{Max}(\{w_n(i)\}_{i=1}^N) \tag{10}$$

where the denominator represents the maximum value of all node weights. The function $P(i)$ represents the probability of view camera $i$ participating in gaussian optimization. This design is noteworthy as it significantly reduces the number of optimization iterations, which accelerates the reconstruction process. Despite this reduction, it effectively enhances the geometric representation of Gaussian points, substantially improving the overall quality of the reconstruction.

## 4 EXPERIMENTS

### 4.1 EXPERIMENTAL SETUP

**Datasets and Metrics.** We evaluate our method on three datasets: Waymo (Sun et al., 2020), KITTI (Geiger et al., 2012), and Mill-19, which includes large scenes like Buildings and Rubble (Turki et al., 2022). For Waymo, we use images from three cameras (front, front-left, front-right) across 32 scenes, totaling about 600 images per scene block. For KITTI, each scene block contains approximately 100 images. Consistent with prior research, we select one out of every eight images

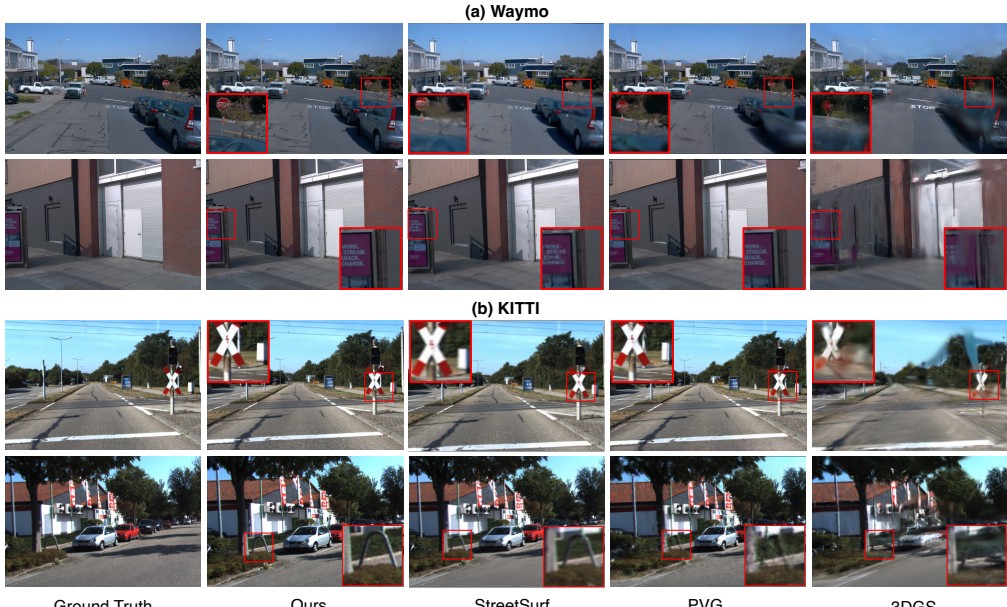

Figure 4: Qualitative comparison of novel view synthesis results on the Waymo and KITTI, showcasing the Ground Truth alongside results from our GraphGS method, StreetSurf (Guo et al., 2023), PVG (Chen et al., 2024), and 3DGS (Kerbl et al., 2023) for comprehensive evaluation. Our approach yields closer fidelity to the Ground Truth, highlighting the effectiveness of our reconstruction method in various urban scene complexities.

| Method | Building | | | Rubble | | |
|--------|----------|--------|---------|--------|--------|---------|
| | PSNR ↑ | SSIM ↑ | LPIPS ↓ | PSNR ↑ | SSIM ↑ | LPIPS ↓ |
| Mega-NeRF (Turki et al., 2022) | 20.93 | 0.547 | 0.349 | 24.05 | 0.553 | 0.373 |
| Switch-NeRF (Mi & Xu, 2023) | 21.54 | 0.579 | 0.294 | 24.31 | 0.562 | 0.329 |
| 3DGS (Kerbl et al., 2023) | 23.01 | 0.769 | 0.164 | 26.78 | 0.800 | 0.161 |
| VastGaussian (Lin et al., 2024) | 23.50 | 0.804 | **0.130** | 26.92 | 0.823 | **0.132** |
| Ours | **26.60** | **0.854** | 0.163 | **27.03** | **0.869** | 0.185 |

Table 2: Quantitative comparison of novel view synthesis on the Mill 19 large scene dataset. for testing, using the remainder for training. Our method, designed for scenarios lacking pose data, does not use ground truth (GT) poses from the datasets.

We compare our method on the Waymo and KITTI datasets against various NeRF-based methods (UC-NeRF (Cheng et al., 2023), Mip-NeRF 360 (Barron et al., 2022), Zip-NeRF (Barron et al., 2023), StreetSurf (Guo et al., 2023), EmerNeRF (Yang et al., 2023a), SNeRF (Xie et al., 2023)) and methods based on 3DGS (3DGS (Kerbl et al., 2023), PVG (Chen et al., 2024)). We specifically evaluate against both pose optimization (UC-NeRF (Cheng et al., 2023)) and pose-free reconstruction methods (e.g., BARF (Lin et al., 2021), SPARF (et al., 2023b), UP-NeRF (et al., 2023a)). For the large Mill-19 dataset, we compare with Mega-NeRF (Turki et al., 2022), Switch-NeRF (Mi & Xu, 2023), and VastGaussian (Lin et al., 2024), segmenting scenes for 3DGS due to memory limits. For some methods, we use published results for comparisons.

**Implementation.** Experiments were conducted using an NVIDIA RTX 3090 GPU and an AMD EPYC 7542 CPU. For image sets lacking pose or sequence information, we use a pre-trained model (Wang et al., 2024) to estimate relative poses in about 0.01 seconds per pair. Although not highly precise, it effectively outlines the coarse distribution of the images. We then apply our scene structure strategy to estimate scene structures, generate camera graphs, and calculate consistency and importance weights. Additionally, we optimize memory and training efficiency by using an octree to prune redundant points from approximately 300k to 100k.

For the quantitative comparison of our Spatial Prior-based Structure Estimation with Colmap, the Colmap setup included a vocabulary tree containing 256K visual words, pre-built using the Flickr100k dataset (available on the COLMAP project page). Further details and training specifics are available in the Appendix.

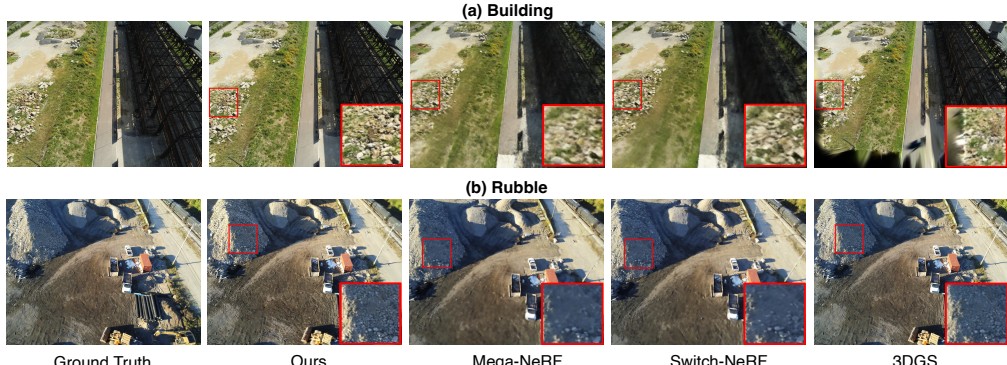

Figure 5: Qualitative comparison of novel view synthesis in the Mill 19 large scene dataset (Turki et al., 2022), showcasing the Ground Truth alongside the results from our method and other state-of-the-art methods including Mega-NeRF (Turki et al., 2022), Switch-NeRF (Mi & Xu, 2023), and 3DGS (Kerbl et al., 2023).

| | Strategy | Matching Time ↓ | BA Time ↓ | PSNR ↑ | SSIM ↑ | LPIPS ↓ |
|---|---|---|---|---|---|---|
| 0.6k | COLMAP (Ex) | 62 min | 140 min | 30.05 | 0.89 | **0.23** |
| | COLMAP (Vo) | 50 min | 104 min | 29.02 | 0.88 | 0.26 |
| | Ours | **3 min** | **20 min** | **30.18** | **0.90** | 0.24 |
| 2k | COLMAP (Ex) | >24 h | | - | - | - |
| | COLMAP (Vo) | >24 h | | - | - | - |
| | Ours | **12 min** | **194 min** | **26.60** | **0.85** | **0.16** |

Table 3: Quantitative comparison of our Spatial Prior-based Structure Estimation for 0.6k and 2k images. At 0.6k, COLMAP-Exhaustive (Ex) and COLMAP-VocabTree (Vo) are benchmarked. At the 2k scene, COLMAP fails consistently, with SfM processing times exceeding 24 hours, yielding no results. Our approach demonstrates significantly faster while maintaining comparable accuracy.

## 4.2 EXPERIMENTAL RESULTS

**Waymo, KITTI and Mill-19.** Table 1 and Figure 4 show that our method outperforms others in both quantitative and qualitative measures, driven by precise structure estimation and graph-guided optimization. It accurately captures fine details such as tree branches without noticeable blurring, while maintaining superior shape, geometry, and color fidelity. Notably, BARF does not converge on two datasets, and SPARF, is suitable only for sparse image sets. UP-NeRF also struggles with convergence on the KITTI dataset. For large scenes like Mill-19, presented in Table 2 and Figure 5, our method continues to demonstrate clear edges and detailed clarity in elements like rubble, grass, and stones.

**Spatial Prior-based Structure Estimation.** We demonstrate the efficiency of our spatial prior-based method through Concentric Nearest Neighbor Pairing (CNNP 3.1.1) and Quadrant Filter (QF 3.1.2). Table 3 shows our method increases matching speed 20x and reduces bundle adjustment time 5x compared to traditional approaches like exhaustive or vocabtree (Schönberger et al., 2016a) matching. For large datasets, where naive 3DGS methods using COLMAP are impractical due to long Structure from Motion times (typically >24 hours and always fails), our method completes structure estimation in just a few hours and achieves higher-quality reconstructions by accurately determining camera poses and avoiding invalid camera pairs.

| Method | PSNR ↑ | SSIM ↑ | LPIPS ↓ |
|---|---|---|---|
| w/o QF | 29.14 | 0.89 | 0.25 |
| w/o CNNP | 24.33 | 0.82 | 0.33 |
| w/o Structure Estimation | 24.77 | 0.803 | 0.444 |
| w/o Multi-view Consistency | 29.42 | 0.834 | 0.297 |
| GraphGS(Ours) | **30.36** | **0.891** | **0.267** |

Table 4: Quantitative Comparison of Ablation Experiments for Submodules.

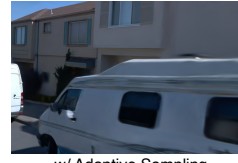 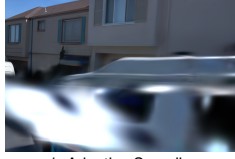 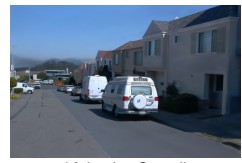 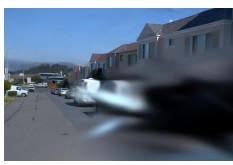

| w/ Adaptive Sampling | w/o Adaptive Sampling | w/ Adaptive Sampling | w/o Adaptive Sampling |

Figure 6: Ablation Experiments on Adaptive Sampling Optimization. This figure highlights the impact of excluding our adaptive sampling optimization.

| Method | Iterations | Training Time ↓ | PSNR ↑ | SSIM ↑ | LPIPS ↓ |
|---|---|---|---|---|---|
| w/o Adaptive Sampling | 30000 | 54 min | 28.81 | 0.884 | 0.253 |
| w Adaptive Sampling | **16830** | **28 min** | **30.36** | **0.891** | **0.267** |

Table 5: Quantitative comparison of Ablation Experiments on Adaptive Sampling Optimization.

### 4.3 ABLATION STUDY

We conducted ablation studies to assess the contributions of our method's components to the reconstruction process. This included evaluations of structural estimation and its sub-modules CNNP and QF, as well as the effects of graph-guided multi-view consistency constraint and adaptive sampling optimization on reconstruction quality.

Table 4 confirms the impact of each module on reconstruction quality. On the Waymo dataset, omitting the QF leads to reduced accuracy due to the inclusion of non-intersecting camera pairs in pose estimation, affecting camera pose precision. Similarly, excluding the CNNP module results in a notable decline in quality because it fails to accurately capture global poses through local matching. Integrating structure estimation significantly enhances reconstruction quality by precisely estimating the global scene structure. Additionally, the inclusion of graph-guided multi-view consistency constraints further improves outcomes by maintaining consistency across multiple views.

Table 5 demonstrates that the adaptive sampling optimization method significantly reduces optimization time and improves the quality of reconstruction. As illustrated in Figure 6, this approach mitigates overfitting to specific viewpoints by optimizing the distribution of Gaussian points. This enhancement not only prevents the occlusion of other perspectives but also reduces severe distortions within the reconstructed geometry. For more ablation experiments, see Appendix E.

## 5 CONCLUSION

In conclusion, through our proposed structure estimation and graph-guided optimization methods, GraphGS can achieve high-quality, rapid reconstruction of large scenes from image sets without ground truth poses. This capability is particularly meaningful for multimedia applications such as virtual reality, gaming, and the metaverse. GraphGS not only meets the growing demand for fast and reliable scene reconstruction but also provides a scalable and accessible solution.

**Limitations.** The absence of rear camera data and the anisotropic nature of the method limit the comprehensive capture and reconstruction of reverse scenes. Additionally, the reliance on feature points may cause the matching strategy to fail randomly, particularly in scenes with numerous dynamic objects. The effects on distant views and the blurring caused by large Gaussian points in certain shots still need improvement and resolution. For more discussion, see Appendix G.

### ACKNOWLEDGMENT

This research is supported by the National Natural Science Foundation of China (No. 62406267), Tencent Rhino-Bird Focused Research Program, Guangzhou-HKUST(GZ) Joint Funding Program (Grant No.2025A03J3956), the Guangzhou Municipal Science and Technology Project (No. 2025A04J4070), the Guangzhou Municipal Education Project (No. 2024312122) and Education Bureau of Guangzhou Municipality.

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

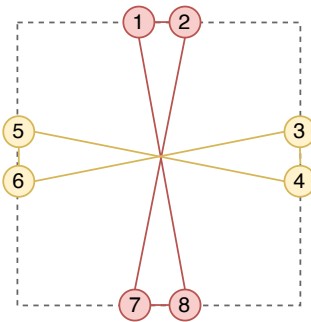

Figure 7: Counter example for one special case of $r$,$h$,$w$. In the special case, $r = 2$, $h = N - 3$, $w = 1$. This means for every camera, we will only add its nearest camera and farthest camera as graph edge. We list one counter example with $N = 8$. In the counter example, the red nodes and yellow nodes are disconnected, while all the nodes follow the matching algorithm.

## A   APPENDIX: PROOF OF GRAPH CONNECTIVITY

For Equation. 1, if we remove $S^{(i)}_{\text{connection}}$, the $S_{all}$ would be as follows:

$$S_{all} = \bigcup_{i=1}^{N} (S^{(i)}_{\text{neighbor}} \cup S^{(i)}_{\text{concentric}}) \tag{11}$$

$$S^{(i)}_{\text{neighbor}} = \{(c_i, c_j)|s(i,j) \leq r, j = 1, 2, ..., N, j \neq i\}$$

$$S^{(i)}_{\text{concentric}} = \{(c_i, c_j)|0 \leq (s(i,j) - r)\%(h + w) < w, j = 1, 2, ..., N, j \neq i\}$$

In the scenario, the camera graph may not be connected graph. It would be fairly troublesome to give direct proof. However, when considering proof by contradiction, one counter example for one special case of $r$,$h$,$w$ is enough.

**Proof:**

We choose the special case when $r = 2$, $h = N - 3$, $w = 1$, which means for every camera node in graph, we only add edges to its nearest camera and farthest camera. In this scenario, one counter example is illustrated in Fig. 7. The graph nodes represent cameras while the position of graph nodes also represents the actual position of cameras. For example, camera node 1 connected camera 2 and camera 8 since they are the nearest and farthest cameras to camera 1 respectively. After the edge adding algorithm done, the camera $1, 2, 7, 8$ and $3, 4, 5, 6$ form two different connected components, while the entire camera graph is not a connected graph.

## B   APPENDIX: STATE TABLE OF QUADRANT FILTER

In this part, we introduce quadrant division and the state table for filtering noise camera pairs as described in Section 3.1.2 of the main paper.

**Quadrant Division.** As illustrated in Figure 8, we adopt the coordinate system of OpenCV, which is a right-handed system. Within this camera coordinate system, it is assumed that the camera is facing $+z$ axis. We adhere to this assumption when calculating the relative orientation of cameras.

**State Table.** For state table, we identify 64 potential configurations describing the relative relationship between two cameras. For the conciseness of representation, we only illustrate the successful matching pairs, as shown in Table 6. The "position quadrant" column lists all possible quadrants of the relative position between cameras $c_i$ and $c_j$, while the "orientation quadrant" column indicates the quadrant of relative orientation that enables successful matching. For instance, if $c_j$ is positioned in the right upper rear quadrant of $c_i$ (quadrant 1 in Fig. 8), only a facing orientation in quadrant 7 (relative orientation) satisfies the matching condition under strict mode. Practically, we employ Loose Mode for vehicle data due to the elongated and narrow camera track. For other situations, we apply Strict Mode.

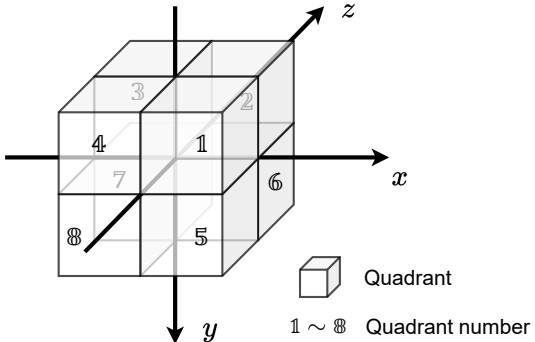

Figure 8: Illustration of quadrant division: When calculating the relative position of cameras $c_i$ and $c_j$, we assume that camera $c_i$ is positioned at the origin. In determining the relative direction, we assume that the orientation of camera $c_i$ is facing the $+z$ axis.

| Position Quadrant | Orientation Quadrant | |
|:---:|:---:|:---:|
| | Strict Mode | Loose Mode |
| 1 | 7 | 2, 3, 6, 7 |
| 2 | 7, 8 | 2, 3, 4, 6, 7, 8 |
| 3 | 5, 6 | 1, 2, 3, 5, 6, 7 |
| 4 | 6 | 2, 3, 6, 7 |
| 5 | 3 | 2, 3, 6, 7 |
| 6 | 3, 4 | 2, 3, 4, 6, 7, 8 |
| 7 | 1, 2 | 1, 2, 3, 5, 6, 7 |
| 8 | 2 | 2, 3, 6, 7 |

Table 6: State Table Illustration.

**Probability of filter under Strict/Loose Mode.** For the situation of random camera position and orientation, the strict and loose mode of state table have different probabilities to filter irrelevant camera pairs, which can be calculated theoretically to measure efficiency. For Strict mode, the probability of filter is

$$P_{\text{strict}} = \frac{7}{8} \times \frac{1}{8} \times 4 + \frac{6}{8} \times \frac{1}{8} \times 4 = \frac{13}{16} \approx 81.3\% \tag{12}$$

For Loose mode, the probability of filter is

$$P_{\text{loose}} = \frac{4}{8} \times \frac{1}{8} \times 4 + \frac{2}{8} \times \frac{1}{8} \times 4 = \frac{6}{16} \approx 37.5\% \tag{13}$$

It is notable even in loose mode, the probability of filter is still a high number, meaning more than $1/3$ noise camera pairs are filtered. This greatly reduced the burden of calculation in the following structure pipeline compared to the situation without quadrant filter.

## C APPENDIX: PROOF OF PROPOSITION

In the Proposition 1 of the main paper, we utilize the internal relations of cross product and inner product within a right-handed 3D coordinate system to calculate the relative orientation without the need to solve the linear system:

$$\mathcal{B}_d^{(i,j)} = \{sgn(e_x[d_\times^{(i)}]d^{T(j)}), sgn(e_y[d_\times^{(i)}]d^{T(j)}), sgn(d^{(i)}d^{T(j)})\} \tag{14}$$

where $[d_\times^{(i)}]$ represents the anti-symmetric matrix of cross product:

$$[d_\times^{(i)}] = \begin{bmatrix} 0 & -v_z^{(i)} & v_y^{(i)} \\ v_z^{(i)} & 0 & -v_x^{(i)} \\ -v_y^{(i)} & v_x^{(i)} & 0 \end{bmatrix} \tag{15}$$

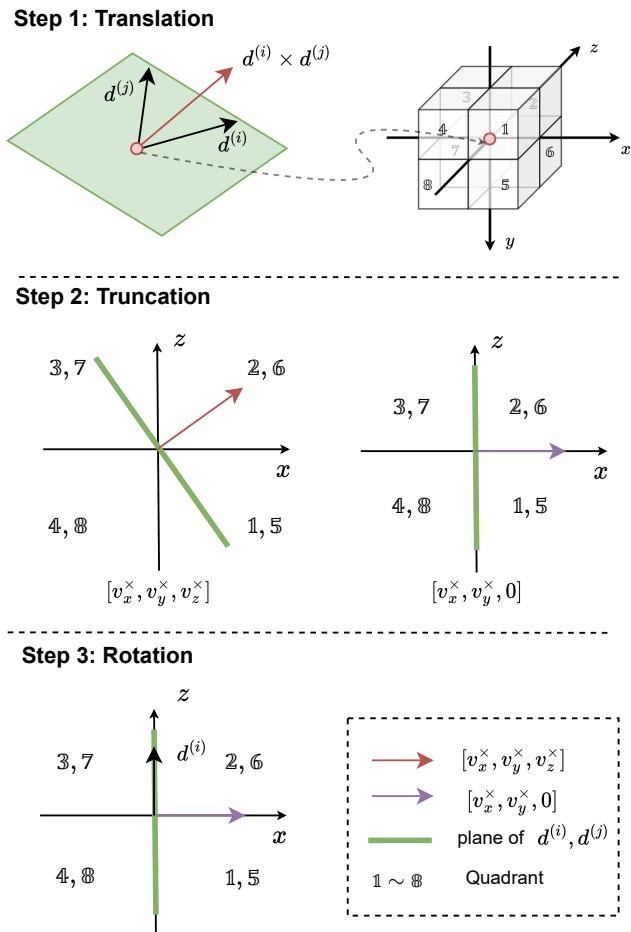

Figure 9: Imagined Transform Illustration: Throughout the entire process, the relative orientation of $d^{(i)}$ and $d^{(j)}$ is maintained without additional calculation.

and $e_x = [1, 0, 0], e_y = [0, 1, 0]$ represent unit vector of $x, y$ axis. However, its mathematical principles are omitted due to limited space. Here we will explain the rationality of our method.

**Proof:**

For cameras $c_i$ and $c_j$ with absolute orientations $d^{(i)}$ and $d^{(j)}$, we use their cross product $d^{(i)} \times d^{(j)} = [v_x^\times, v_y^\times, v_z^\times]$ to reflect the relative orientation to some extent. We envision transforming them into a standard coordinate system as illustrated in Figure 8 through several steps:

- **Step 1:** Translate them to the origin of the standard coordinate. This process does not alter the cross product or their relative orientation.

- **Step 2:** From an overhead view of the standard coordinate (where the $y$-axis points into the page), we note that the plane formed by $d^{(i)}$ and $d^{(j)}$ divides the $x$-$z$ plane into two parts. We then transform $d^{(i)}$ and $d^{(j)}$ such that $d^{(i)} \times d^{(j)}$ changes from $[v_x^\times, v_y^\times, v_z^\times]$ to $[v_x^\times, v_y^\times, 0]$, while maintaining their relative orientation. This transformation is feasible because the linear system we aim to solve is not full rank. After this transformation, the plane of $d^{(i)}$ and $d^{(j)}$ coincides with the $z$-$y$ plane.

- **Step 3:** We rotate $d^{(i)}$ and $d^{(j)}$ around the $x$-axis so that $d^{(i)}$ aligns with the $z$-axis.

After the aforementioned steps, as depicted in Figure 9, we have successfully translated $d^{(i)}$ and $d^{(j)}$ from an arbitrary situation to the standard coordinate with $d^{(i)}$ pointing toward the $+z$ axis. In this

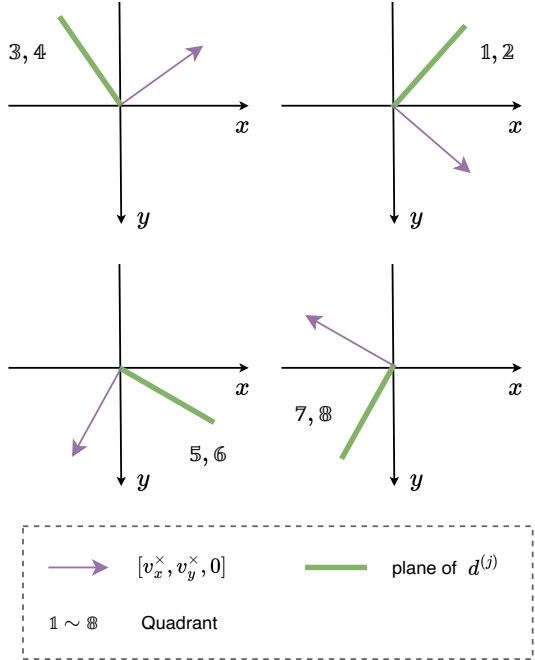

Figure 10: Four situations for the plane of $d^{(j)}$, which depends on the sign of $v_x^\times, v_y^\times$.

process, no transformation matrix is required, while $v_x^\times$, $v_y^\times$, and the relative orientation of $d^{(i)}$ and $d^{(j)}$ are preserved. This sets the stage for further analysis.

We then consider all potential positions of $d^{(j)}$. As shown in Figure 10, $d^{(j)}$ lies on a green plane that can rotate around the $z$-axis. Viewing the coordinate system from the front, where the $z$-axis points into the page, and due to the constraints of the right-handed rule, the potential positions of $d^{(j)}$ are limited to half of the green plane, contingent upon one of four scenarios based on $[v_x^\times, v_y^\times, 0]$. Thus, we can limit the relative orientation of $d^{(j)}$ to two of the eight quadrants through $v_x^\times$ and $v_y^\times$.

Finally, we simply calculate the inner product of $d^{(i)}$ and $d^{(j)}$ to determine the relative orientation along the $z$-axis (either quadrant $2, 3, 6, 7$ or $1, 4, 5, 8$). Based on the derivation above, the final $\mathcal{B}_d^{(i,j)}$ can be rewritten in a more direct form without a matrix:

$$\mathcal{B}_d^{(i,j)} = \{\text{sgn}(v_x^\times), \text{sgn}(v_y^\times), \text{sgn}(d^{(i)} \cdot d^{(j)})\} \tag{16}$$

The Equation 16 and Equation 14 are actually equivalent, while we keep Equation 14 in the main text for the consistency of context.

## D   APPENDIX: ADDITIONAL EXPERIMENTAL RESULTS

| Method | Time (min) | PSNR | SSIM | LPIPS |
|---|---|---|---|---|
| ACEZero | 10 | 17.50 | 0.725 | 0.354 |
| PixSfM | 130 | 28.75 | 0.847 | 0.366 |
| GLOMAP | 28 | 27.65 | 0.824 | 0.388 |
| COLMAP | 154 | 29.14 | 0.89 | 0.250 |
| **Ours** | **23** | **30.36** | **0.891** | **0.267** |

Table 7: Quantitative Comparison of SfM on Waymo dataset.

As shown in Table D, our method demonstrates significant improvements over other state-of-the-art SfM systems in terms of processing time and accuracy metrics. Despite the emergence of numerous

| Method | Initial Points Number | Training Time ↓ | PSNR ↑ | SSIM ↑ | LPIPS ↓ |
|---|---|---|---|---|---|
| w/o Octree Point Initialization | 245820 | 54 min | 29.74 | 0.841 | **0.291** |
| w Octree Point Initialization | 100000 | **36 min** | **29.75** | **0.843** | 0.292 |

Table 8: Quantitative comparison of Ablation Experiments on Octree Point Initialization.

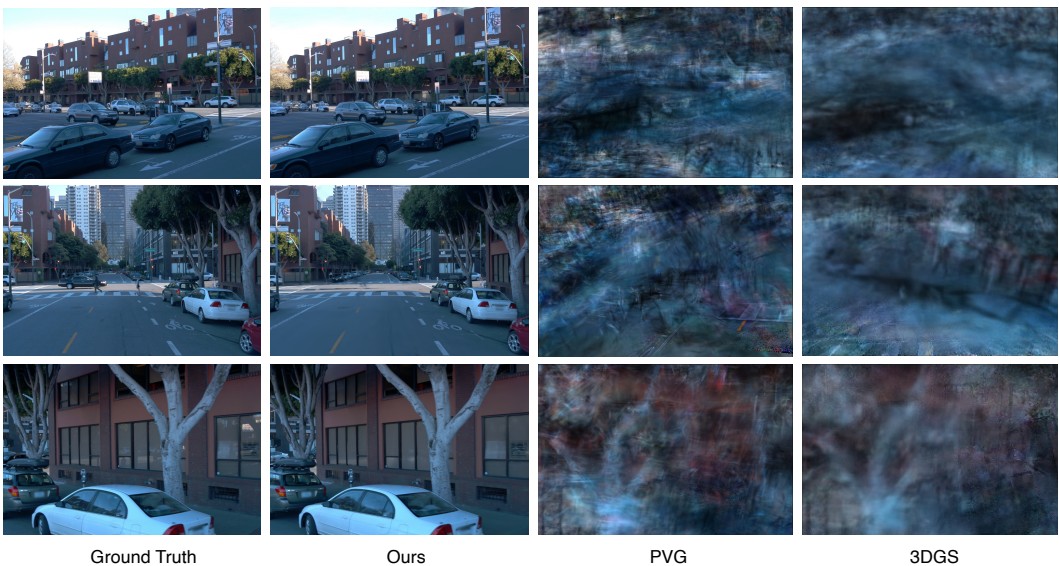

Ground Truth      Ours      PVG      3DGS

Figure 11: Qualitative comparison of novel view synthesis with added random Gaussian noise to simulate imprecise pose data. Our method retains clarity and definition, while alternative approaches like PVG (Chen et al., 2024) and 3DGS (Wu et al., 2023) show distortions under similar conditions.

effective Structure from Motion (SfM) methods, COLMAP continues to be widely recognized for its robustness and accuracy in the community, thereby serving as a benchmark for comparisons among incremental SfM (Pan et al., 2024). Notably, VGGSfM (Wang et al., 2023b) was excluded from detailed comparisons due to memory limitations when processing large image datasets. In contrast, ACEZero Brachmann et al. (2024), although faster, faces convergence issues in larger datasets. Our method not only surpasses others in terms of results from novel view synthesis but also maintains a faster processing speed.

## E APPENDIX: ADDITIONAL ABLATION STUDIES

**Impact of Octree Initialization on reconstruction quality and training time.** Quantitative results in Table 8 show that our octree point initialization strategy cuts training time in half without compromising reconstruction quality. Further reductions in initial points did not significantly decrease training times, likely due to the densification and point operations.

**Camera Disturbance.** To simulate real-world conditions where camera poses are uncertain, we introduce disturbances by adding random noise from $0$ to $0.3$ radians to camera orientations and positions in a selected Waymo scene. We assess the robustness of our methods by comparing them with 3DGS and PVG under these conditions.

Our method remains robust against disturbances in camera poses, common in real-world scenarios like road bumps. Figure 11 and Table 9 show that traditional 3DGS methods degrade significantly

| Method | PSNR ↑ | SSIM ↑ | LPIPS ↓ |
|---|---|---|---|
| 3DGS ($\mathbf{R}$*0.3 & $\mathbf{T}$*0.3) | 12.676 | 0.510 | 0.613 |
| PVG ($\mathbf{R}$*0.3 & $\mathbf{T}$*0.3) | 12.695 | 0.512 | 0.615 |
| Our ($\mathbf{R}$*0.3 & $\mathbf{T}$*0.3) | **28.53** | **0.886** | **0.221** |

Table 9: Quantitative comparison of novel view synthesis results on Waymo dataset with additional noise. $\mathbf{R}$*0.3 and $\mathbf{T}$*0.3 represent the addition of 0.3 units of rotation and translation noise, respectively.

| Images | Method | Pairing Time | Matching Time | BA Time | PSNR | SSIM | LPIPS |
|---|---|---|---|---|---|---|---|
| 0.6 k | Ours | 0.2 min | 3 min | 20 min | 30.18 | 0.90 | 0.24 |
| 0.6 k | Frusta | 2 min | 45 min | 86 min | 30.15 | 0.89 | 0.24 |
| 2 k | Ours | 2.1 min | 12 min | 194 min | 26.60 | 0.85 | 0.16 |
| 2 k | Frusta | 20 min | >24 h | >24 h | - | - | - |

Table 10: Comparation of our Concentric Nearest Neighbor Pairing (CNNP), Quadrant Fileter (QF) and naive frusta method on Waymo (0.6 k) and Mill 19 (2 k) datasets.

under such disturbances, with heavy blurring that obscures scene objects. In contrast, our approach maintains quality by calibrating camera poses using key camera pairs.

**Comparison with naive overlapping frusta method** We conduct an experiment with the naive frusta method and compare it to our Concentric Nearest Neighbor Pairing (CNNP), Quadrant Fileter (QF) methods when selecting matched pairs. We implement the frusta algorithm with the following steps:

- The intrinsic matrix of camera is obtained to calculate the near-far plane.
- The frusta of cameras is transformed to world coordinate system.
- We check the intersection of their bounding box to judge the view intersection.

As shown in Tab. 10, and further analysis are presented as follows:

- **Universality**. Notably, our matching strategy only needs the position and orientation of coarse cameras, i.e., only the 3-rd, 4-nd colume of extrinsic matrix. Moreover, we do not need the information of intrinsic matrix. This indicates the universality of our method.
- **Efficiency**. As shown in QF section and Appendix C, we calculate the relative orientation with only 1 inner product and 1 cross-product based on fully convincing mathematical proof. As shown in column of pairing time, this way greatly improves our efficiency compared to frusta method, which contains time-consuming coordinate transform and intersection checking process.
- **Reconstruction Quality**. In terms of reconstruction quality, there is no obvious difference between our method and frusta, since pose accuracy is well-solved in BA procedure. Our approach offers better compatibility in cases where initial poses are poor.
- **Reason why frusta failed**. Though the naive frusta method sounds reasonable, however, in practice, especially in the driving scenario with long and narrow camera track, the frusta method cannot filter image pairs efficiently since all of the camera orientation are extremely similar.

## F   APPENDIX: EXPERIMENTAL DETAILS

Experiments were conducted using an NVIDIA RTX 3090 GPU and an AMD EPYC 7542 CPU. The Waymo includes 600 images from three viewpoints (left, center, right), covering 599 image

pairs. The Kitti includes 100 images with 99 image pairs. The Mill 19 comprises 2000 images with 1999 pairs included.

For the CNNP configuration, the settings of $r = 5$, $h = 20$, $w = 1$ were employed. Under this setting, approximately $\sim 18k/C_{600}^2$ pairs will be selected for 0.6k images, and $\sim 200k/C_{2000}^2$ pairs will be selected for 2k images. The setting $r = 5$ designates the five nearest cameras as matching candidates for each camera $c_i$. The configuration $h = 20$, $w = 1$ means that one camera is chosen as a match for $c_i$ out of every 20 cameras based on proximity.

To optimize multi-view consistency and balance the coefficients, we set the coefficient $\lambda$ to 0.07, which was empirically found to be optimal in our experiments. Additionally, we adopted a graph-guided optimization setup, where the sampling probability is determined by the weights of the graph nodes. We set the minimum sampling probability at 0.5 to ensure that nodes with lower weights are not overlooked.

## G    APPENDIX: DISCUSSION OF LIMITATIONS AND CHALLENGES.

Our research focuses on 3DGS reconstruction of outdoor, unbounded scenes, primarily facing two challenges: 1) Accurate pose estimation outdoors is difficult due to the unpredictability and complexity of outdoor environments; 2) Outdoor scenes generally have sparser camera coverage and less overlap between images compared to indoor settings, resulting in insufficient constraints during training. To address these challenges, we introduced two key modules: Spatial Prior-Based Structure Estimation and a graph-guided Gaussian optimization strategy. These modules are designed to efficiently and accurately complete the reconstruction of outdoor scenes.

However, our method shows limited improvement in 3DGS reconstruction of objects. This is primarily because datasets in this category usually have precise ground truth poses, and the camera setups often rotate 360 degrees around the object, providing sufficient image overlap and constraints to aid convergence. This results in nearly equal importance weights for graph nodes, which diminishes the impact of our optimization.

Moreover, our approach relies on the accuracy of rough spatial priors. If the initial camera pose distribution in the xy plane is inaccurate, it may lead to Bundle Adjustment (BA) failure. In such cases, our method might not effectively handle pose estimation errors.

