# OpenReview forum: "Graph-Guided Scene Reconstruction from Images with 3D Gaussian Splatting"
_ICLR.cc/2025/Conference — ICLR 2025 Poster_

### Official Review · Reviewer_X4qU · 2024-10-27

**Soundness:** 3
**Presentation:** 3
**Contribution:** 3
**Rating:** 6
**Confidence:** 5

**Summary:**

The authors propose a framework for large-scale 3D reconstruction using Gaussian splatting.  The authors suggest the construction of a prior graph-guided scene structure. This results in estimating a camera graph that encodes the camera topology. Then based on graph weights the employ an adaptive sampling strategy to the 3D Gaussian splatting optimization.

**Strengths:**

The authors present an elaborate pre-processing framework to effectively scale Gaussian splitting to large scenes. The ideas presented in the paper are exciting and well-presented for the most part. The concept of exploiting low-cost prior heuristics to allow the network to focus on the underlying task is interesting and the experimental evaluation demonstrates the effectiveness of the method.

**Weaknesses:**

1) minor: There are some spelling errors, please proofread the manuscript e.g ( line 135 formed -> form, line 138 initializaition -> initialization )

2) Line 147 - 149: Please specify here which models you use to obtain camera poses. How approximate are these poses? What is the error tolerance? Could you please provide quantitative metrics on the initial pose quality compared to ground truth if available?

 3) For the concentric nn pairing the authors use the symbol S multiple times. It would make sense to use CNNR as a symbol of the overall process output and use more meaningful names than S1 S2 and S3 for the various heuristics. Maybe names related to their role in CNNR computation.

4) In the quadrant filer could you please specify whether the orientation is provided as a normal or any other form (euler angles)?

5) The adaptive sampling section is not clear.
    - line 322 primarily considers two criteria -> Are there more criteria than these two?
    - Line 344 We design node weight wn(i) based on betweenness centrality -> So the node weight does not take into account the degree centrality?
    - How is the view selection probability integrated into the 3DGS optimization?

6) Lines 457-458: Could you please provide more information on how you obtained your initial poses and what is the size of the dataset, the hardware used, or any preprocessing stats that the relative pose estimation is done in 10ms?

**Questions:**

What is the minimum quality of poses required by the proposed framework?
Does any of the proposed steps account for large errors in the pose estimation?
Why do the authors attempt to compete strongly with COLMAP? COLMAP was used as a method to obtain initial poses and the authors used wang et al. 2024 to obtain initial poses. The framework the authors propose can be used with poses computed by either method.

---

> ### Author Response · Authors · 2024-11-21
>
> Thank you for recognizing our ideas, presentation, and experiments.
> > **Q1: Spelling Errors.**
>
> **A:** Thank you for pointing them out; we will carefully make the corrections.
>
> > **Q2: Details on acquiring initial camera poses.**
>
> **A:** We use only Dust3R’s pairwise pose estimation module without the Global Alignment module to obtain rough initial poses. Each image pair takes approximately 0.01 seconds on the NVIDIA A6000, with 599 pairs from 600 images in the Waymo dataset and 1999 pairs from 2000 images in the Mill 19 dataset. Given that the dataset’s initial poses are less accurate than our results, we compare them against COLMAP benchmarks and our final poses in Waymo, as the table below:
> |                     | ATE RMSE         |    PSNR           |    SSIM           |   LPIPS           |
> |:-----------:        |----------        |:---------:        |:---------:        |:---------:        |
> | Init. Pose          |   2.13           |   23.67           |   0.785           |   0.319           |
> | Final Pose         |   0.04           | **30.36**         | **0.891**         | **0.267**         |
>
> > **Q3: Symbol for concentric nearest neighbor pairing.**
>
> **A:** Thanks for pointing this out, we will rename S_1, S_2, and S_3. S_1 is related to the nearest neighbor cameras, so we rename it to S_neighbor; S_2 are the cameras with isometric interval, so we rename it to S_concentric; S_3 is designed for connection of camera graph, so we rename it to S_connection.
>
> > **Q4: Quadrant filter details.**
>
> **A:** The orientation is a 3D vector which is equivalent to the 3-rd column of the extrinsic matrix. This provides convenience for the following cross-product and inner-product.
>
> > **Q5: Details of adaptive sampling.**
>
> **A:** Degree centrality focuses on the activity level of nodes, while betweenness centrality is concerned with the control a node exerts or its role as a bridge. In our experiments, we found that using only betweenness centrality yielded better results, so we only use betweenness centrality to design node weights.
>
> > **Q6: Details on how to obtain the initial pose.**
>
> **A:** The Details as below:
> 1. Time and Hardware: Each pair is estimated in 0.01 seconds using an NVIDIA RTX3090 with 24GB of memory. There is no need for Dust3R’s global alignment, as it relies solely on pairwise pose estimation.
> 2. Datasets: For the Waymo dataset, we use 599 pairs (600 images) for evaluation; For the Mill 19 dataset, we use 1999 pairs (2000 images) to do evaluation.
>
> Please be clarified that Dust3R is not mandatory. Coarse poses from the datasets can also be used directly in our proposed framework. For initial pose accuracy evaluations, please refer to Q2.

---

> ### Author Response · Authors · 2024-11-21
>
> > **Q7: The minimum pose quality, the error tolerance, and the reasons for comparison with COLMAP.**
>
> **Q7.1: What quality of poses does the framework require?**
>
> **A:** Our goal is to develop a framework for reconstructing outdoor scenes from uncalibrated images. We use coarse pose prior to speed up the Structure-from-Motion (SfM) process and build a graph-guided 3DGS reconstruction. The Dust3R for obtaining rough poses is optional; any coarse pose is suitable, but we choose it for its rapid and widely recognized capabilities. As long as the initial results with correct camera layout on the xy-plane can be used for successful bundle adjustment (BA), we regard them to meet the minimum quality for initial poses. For initial pose quality quantitative comparison, please see Q2.
>
> **Q7.2: Are there measures to handle large pose errors?**
>
> **A:** Our matching strategy requires only coarse information about the camera’s position and orientation, specifically the third and fourth columns of the extrinsic matrix. This provides a higher error tolerance; as long as the initial results do not produce severe camera layout errors on the xy-plane, BA can be completed.
> If BA proceeds successfully, it will not affect subsequent accuracy estimations. However, how to handle BA failures is beyond the scope of this paper. In the test datasets we used, BA is successfully completed.
>
> **Q7.3: Why does it compete with COLMAP?**
>
> **A:** The comparison with COLMAP is due to its wide recognition as an SfM benchmark in the community.
> The latest SfM methods like VGGSfM [R1], GLOMAP [R2], and ACEZero [R3] all use COLMAP as their primary comparative baseline. We have extensively evaluated these recent methods on the Waymo dataset.
> | Methods         | Time            |    PSNR           |    SSIM           |   LPIPS           |
> |:-----------        |---------        |:---------:        |:---------:        |:---------:        |
> | VGGSfM[R1]          | -               | -                 | -                 | -                 |
> | Ace0 [R3]            | **10min**           |   17.50           |   0.725           |   0.354           |
> | PixSfM [R4]          | 130min          |   28.75           |   0.847           |   0.366           |
> | Glomap [R2]         | 28 min         |   27.65           |   0.824           |   0.388           |
> | Colmap          | 154min          |   29.14           |   0.89           |   0.250           |
> | Ours            | 23 min          | **30.36**         | **0.891**         | **0.267**         |
>
> As shown in the table, VGGSfM encounters an out-of-memory (OOM) issue when processing 600 images, ACEZero is fast but fails to converge entirely, and other methods also underperform in pose performance for novel view synthesis compared to our method. Therefore, COLMAP remains the most competitive benchmark for comparison.
>
> **Reference**
>
> [R1] J. Wang et al., “Visual Geometry Grounded Deep Structure from Motion,” arXiv preprint arXiv:2312.04563, 2023. [Online]. Available: https://arxiv.org/abs/2312.04563
>
> [R2] L. Pan et al., “Global Structure-from-Motion Revisited,” arXiv preprint arXiv:2407.20219, 2024. [Online]. Available: https://arxiv.org/abs/2407.20219
>
> [R3] E. Brachmann et al., “Scene Coordinate Reconstruction: Posing of Image Collections via Incremental Learning of a Relocalizer,” arXiv preprint arXiv:2404.14351, 2024. [Online]. Available: https://arxiv.org/abs/2404.14351
>
> [R4] P. Lindenberger et al., “Pixel-Perfect Structure-from-Motion with Featuremetric Refinement,” arXiv preprint arXiv:2108.08291, 2021. [Online]. Available: https://arxiv.org/abs/2108.08291

---

### Official Review · Reviewer_MKvT · 2024-11-04

**Soundness:** 3
**Presentation:** 3
**Contribution:** 2
**Rating:** 6
**Confidence:** 3

**Summary:**

Focus on the large-scale Gaussian-based reconstruction pipeline, the authors propose a graph-guided framework, GraphGS, which leverages spatial priors for scene structure estimation to create a camera graph encoding camera topology. Using graph-guided multi-view consistency and an adaptive sampling strategy, GraphGS enhances the 3D Gaussian Splatting optimization, mitigating overfitting to sparse viewpoints and accelerating reconstruction. Quantitative and qualitative evaluations across multiple datasets demonstrate that GraphGS achieves state-of-the-art performance in 3D reconstruction.

**Strengths:**

1. the paper is easy to understand.
2. the results show the effectiveness of proposed pipeline.

**Weaknesses:**

1. The manuscript lacks a detailed explanation for each term in Equation (1), which would enhance clarity and understanding.
2. Many of the authors' methods are designed to improve upon COLMAP. It would be beneficial to include experiments comparing the accuracy of initial values in GS, such as pose accuracy, to illustrate the improvements.

**Questions:**

1. If the spatial prior-based structure relies on the initial pose estimation, wouldn’t inaccurate initial results lead to a poorly constructed graph?
2. In Table 1, why doesn’t the FPS of the proposed method exceed that of the original 3D Gaussian Splatting (3DGS)? Intuitively, the pose estimation should contribute positively. Where is the additional computation time being spent?

---

> ### Author Response · Authors · 2024-11-21
>
> Thank you for recognizing the clarity of our paper and the effectiveness of our methods.
>
> > **Q1: A detailed explanation of Equation (1).**
>
> **A:** Thanks for your suggestion, we will add more detailed explanation in the revision. In Equation 1, we traverse all of the cameras (from 1 to N, supposing the current camera as c_i) and calculate which camera c_j should form a matching pair with c_i. We have the following steps:
> 1. For c_i, we sort other cameras based on their distance to c_i
> 2. We add c_i 's nearest r cameras, as S_1
> 3. We add w cameras from every h+w camera based on the same sorted order, as S_2.
> 4. We add c_(i-1) in the main loop to match c_i, as S_3
>
> > **Q2: Compare the accuracy of the initial values.**
>
> **A:** We provide initial pose accuracy evaluations in Waymo in the table below:
>
> |   Methods           | ATE RMSE         |    PSNR           |    SSIM           |   LPIPS           |
> |:-----------        |----------        |:---------:        |:---------:        |:---------:        |
> | Init. Pose          | 2.13             |   23.67           |   0.785           |   0.319           |
> | Final. Pose         | 0.04             | **30.36**         | **0.891**         | **0.267**         |
>
> Our previous experiments indicate that the true poses provided by the dataset perform poorly. Therefore, we compare both the initial poses and the final results optimized by our pose estimation module against the COLMAP results as a benchmark. This comparison validated our method’s ability to significantly enhance accuracy.
>
> > **Q3: Tolerance of Spatial prior-based structure and graph construction.**
>
> **A:** Sure, inaccurate initial results do lead to poorly constructed graph. However, our matching strategy requires only coarse information about the camera’s position and orientation, specifically the third and fourth columns of the extrinsic matrix. This offers a higher tolerance for errors; as long as the initial results with correct camera layout on the xy-plane can be used for successful bundle adjustment (BA), we regard them to meet the minimum quality for initial poses.
>
> If BA succeeds, it means the camera graph has already passed the BA test, and the graph topology has no big problem. However, how to handle BA failures is beyond the scope of this paper.
>
> > **Q4: Comparison of FPS with 3DGS.**
>
> **A:** In our experiments, FPS refers solely to the rendering speed after training completion, with 3DGS and our results remaining within a reasonable fluctuation range. The pose estimation method you mentioned affects the overall training time, which should be compared with the 3DGS + Colmap pipeline. As shown in the table below, we significantly lead in both efficiency and quality in Waymo:
> | Images         |    Methods            | SfM Time         | Training Time         | Total Time         | PSNR          | SSIM          | LPIPS         |
> |--------        |:-------------        |:--------:        |:-------------:        |:----------:        |-------        |-------        |-------        |
> |  0.6k          | Colmap + 3DGS         |  154min          |     54min             |   208min           | 28.81         | 0.884         | 0.253         |
> |  0.6k              | Ours                  |   23min          |     28min             |    51min           | 30.36         | 0.891         | 0.267         |

---

### Official Review · Reviewer_4dAT · 2024-11-04

**Soundness:** 3
**Presentation:** 3
**Contribution:** 3
**Rating:** 6
**Confidence:** 4

**Summary:**

The paper presents a collection of practical optimizations to improve quality and efficiency of Gaussian Splatting reconstructions from image collections without poses. The optimizations relate to
1. Efficient View-Pair Finding for Match-graph Construction for Structure from Motion
2. Octree initialization of 3D points and Level-of-details based pruning
3. Multi-view Consistency Loss in 3DGS optimization
4. Match-graph / Camera-graph Importance based View Sampling for 3DGS optimization

The author show results indicating,
Optimization 1 leads to faster SfM (Table 3) and quality improvements in GS (Table 4)
Optimizations 2, 4 leads to faster GS optimization (Table 5, Table 7)
Optimizations 3, 4 leads to quality improvements in GS (Table 4, Table 7)

The results are evaluated on scenes from Waymo, Kitti, and Mill-19 datasets with images in the range of ~600, ~100, and ~2000.

**Strengths:**

Overall, I like that the authors propose very practical optimizations that bring both quality and run-time improvements.

* Originality: The key original contribution of the paper is: Concentric Nearest Neighbor Pairing (CNNP) and Quadrant Filter (QF) organizing/pruning view-pairs in camera-graph. Other contributions are good practical applications of previously known ideas.
* Clarity: The paper is written in a clear language, structured well, and shows experimental validation of the proposed ideas.
* Significance: The paper introduces practical ideas for 3DGS from no-pose image collections.

**Weaknesses:**

The method proposed in the paper is forming camera-graph using Dust3r to find relative poses between image pairs and pruning pairs using the proposed CNNP and QF steps. The efficiency of structure estimation is estimated w.r.t default COLMAP pipeline (assuming incremental Structure from Motion). This is not a fair comparison and sufficient details are not provided, making it harder to assess the true benefit of the proposed improvements.

- Paper mentions that Dust3r is used to estimate pairwise relative poses in 0.01 seconds. Can you provide a more detailed breakdown of the Dust3r usage, including whether the 0.01 seconds is per pair or total, what specific hardware was used, and how many total pairs were evaluated across the different datasets

- COLMAP exhaustive and COLMAP vocab-tree matching time are provided. However, sufficient details on the experimental setup and compute resources are not provided. For example, for vocab-tree matching, which dataset is used to compute the vocab-tree, how many nearest neighbors are retrieved per image, in total how many pairs are evaluated? What compute resources are used for this matching?

Without these details, it is difficult to draw conclusions.

At a more basic level, Dust3r + CNNP + QF contributions are mainly to improve match-graph construction and BA runtime. A fair comparison of the improvements in these runtimes should be with other SoTA efficient SfM methods, not default COLMAP.

Default incremental SfM implemented with COLMAP is commonly used by radiance field papers to compute poses but by no means this is the most efficient pipeline. There is a vast literature on how to approximate match-graph construction going back a decade. There are well-established alternatives to incremental Structure from Motion with implementations in Open source SfM libraries such as OpenMVG, OpenSfM, Theia, and most recently GLOMAP that offer much better run-time behavior.

Given the authors use prior poses estimated from Dust3r, the comparison of CNNP and QF steps should be done with match-graph pruning methods that already use prior poses. A naive baseline to compare against would be to construct a camera-graph only from view pairs with overlapping frusta. Can you add this comparison to your evaluation, evaluating both quality and run-time?

I like the practicality of proposed ideas but I don't think that they are contextualized and compared correctly. The other ideas such as LOD-based point pruning and view-importance based sampling are nice practical improvements which provide qualitative and runtime gains w.r.t. original 3DGS paper, however 3DGS provides a baseline not SoTA comparison. A number of methods have been proposed since the original paper to improve both, the quality and efficiency of 3DGS (2DGS, RadSplats, . A few relevant to the paper: Hierarchical 3D Gaussian Representation (Kerbl et al SIGGRAPH Asia 2024), Scaffold-GS (Lu et al CVPR 2024), Octree-GS (Ren et al).

The authors can also provide results on MipNeRF360 dataset which is used more commonly in radiance field literature, this will make it easier to compare their results against contemporary 3DGS methods.

As is, the paper is an assortment of good practical improvements for a sparse recon + 3DGS reconstruction system, and I am positive that these insights can be valuable for practitioners in the field. However, these small contributions are scattered across the pipeline and none are evaluated as thoroughly as they should be with SoTA methods and good baselines respectively for each, making it difficult to place the value/significance of contributions in context of SoTA.

**Questions:**

- See questions in Weaknesses sections.
- What does "w/o structure estimation" mean in Table 4.
- Which datasets are used for results in Table 4, Table 5, Table 7.
- Is Dust3r used for only pairwise pose estimation or is the step that yields globally aligned point maps and poses also used?
- Definition of S_3^i in Eqn 1. is ambiguous, can you clarify what does this set include?

Minor :
- Abstract mentions that : "This paper investigates ...  reconstructing high-quality, large-scale 3D open scenes from images." but the paper deals with scenes with 100 images, 600 images, and two scenes with 1500-2000 images. Typically large-scale in context of SfM and 3DGS refers to city-scale scenes with tens of thousands of images.
- It should be clarified if I(p) in Eqn 8 means "intensity or color at pixel p in the image" refers to GT image or rendered image.

---

> ### Author Response · Authors · 2024-11-21
>
> Thank you for your detailed feedback. We appreciate your recognition of the practicality of our method. To our best knowledge, we are the first to propose a graph-guided 3DGS optimization method for outdoor scenes, which can effectively accelerate and enhance the performance of 3DGS methods such as 3DGS, OctreeGS, ScaffoldGS, 2DGS, etc.
>
> > **Q1: Implementation Details of Dust3R.**
>
> **A:** The Details as below:
> 1. Time and Hardware: Each pair is estimated in 0.01 seconds using an NVIDIA RTX3090 with 24GB of memory. There is no need for Dust3R’s global alignment, as it relies solely on pairwise pose estimation.
> 2. Datasets: For the Waymo dataset, we use 599 pairs (600 images) for evaluation; For Mill 19 dataset, we use 1999 pairs (2000 images) to do evaluation.
> Please be clarified that Dust3R is not mandatory. Coarse poses from the datasets can also be used directly in our proposed framework.
>
> > **Q2: Experiment Details for COLMAP.**
>
> **Q2.1: Which dataset is used to compute the vocab-tree？**
>
> **A:** Dataset Settings: We use the Waymo dataset, which contains 600 images, selecting three views (left, middle, right) for each. We also use the Building dataset from Mill-19, which contains 2000 images.
> The setting of vocab-tree: The vocab tree is pre-build on the Flickr100k dataset, which is provided officially on https://demuc.de/colmap/ . We use the item "Vocabulary tree with 256K visual words" for evaluation.
>
> **Q2.2: How many nearest neighbors are retrieved, and how many pairs are evaluated in total?**
>
> **A:** The setting of CNNP is "r=5, h=20, w=1", which is an empirical setting with a lower failure rate of BA. Under this setting, ~ 18k / $C_{600}^2$ pairs will be selected for 0.6k images; ~ 200k / $C_{2000}^2$ pairs will be selected for 2k images.
> Explanation of parameters of CNNP:
> 1. "r=5" represents for every camera c_i, the nearest 5 cameras to c_i will be selected as matching pairs
> 2. "h=20, w=1": For camera c_i, we select 1 camera from every 21 cameras as matching pairs. This selection is based on distance order to c_i too. As illustrated in the main paper Fig 2. (left)
>
> **Q2.3: Computing Resources.**
>
> **A:** On GPU NVIDIA RTX3090 and CPU AMD EPYC 7542, COLMAP’s SfM time is 104 minutes for the Waymo dataset and exceeds 24 hours for the Mill 19 dataset. In contrast, our method’s SfM time is 23 minutes for the Waymo and 206 minutes for the Mill 19.
>
> > **Q3: Comparison with Other Existing Efficient SfM Methods.**
>
> **A:** Despite numerous efficient SfM methods being proposed, COLMAP remains widely recognized by the community and is considered a crucial benchmark for comparison. As mentioned in GLOMAP [R2], “GLOMAP achieves a similar level of robustness and accuracy as state-of-the-art incremental SfM systems (COLMAP, Schönberger & Frahm) while maintaining the efficiency of global SfM pipelines.”
> Moreover, the latest SfM methods like VGGSfM [R1], GLOMAP [R2], and ACEZero [R3] also use COLMAP as their primary comparative baseline. We have extensively evaluated these recent methods on the Waymo dataset.
> | Methods         | Time            |    PSNR           |    SSIM           |   LPIPS           |
> |:---------:        |---------        |:---------:        |:---------:        |:---------:        |
> | VGGSfM[R1]          | -               | -                 | -                 | -                 |
> | Ace0 [R3]            | **10min**           |   17.50           |   0.725           |   0.354           |
> | PixSfM [R4]          | 130min          |   28.75           |   0.847           |   0.366           |
> | Glomap [R2]         | 28 min         |   27.65           |   0.824           |   0.388           |
> | Colmap          | 154min          |   29.14           |   0.89           |   0.250           |
> | Ours            | 23 min          | **30.36**         | **0.891**         | **0.267**         |
>
> As shown in the table, VGGSfM encounters an out-of-memory (OOM) issue when processing 600 images, ACEZero is fast but fails to converge entirely, and other methods also underperform in pose performance for novel view synthesis compared to our method. Due to the earlier publication dates of OpenMVG, OpenSfM, and TheiaSfM that the reviewer mentioned, along with the time constraints of the rebuttal, we focus our comparison on the latest GLOMAP and other recent methods for this stage. More results will be added in a future version later.

---

> > ### Author Response · Authors · 2024-11-21
> >
> > > **Q4: Comparison with naive overlapping frusta.**
> >
> > **A:** Thanks for your suggestion, we list the results of our method compared to the naive overlapping frusta method.
> > | Images | Method        | Pairs computing T | Matching Time | BA Time | PSNR | SSIM | LPIPS |
> > |--------|---------------|-------------------|---------------|---------|------|------|-------|
> > | 0.6 k  | ours (CNNP+QF)| 0.2 min           | 3 min         | 20 min  | 30.18| 0.9  | 0.24  |
> > | 0.6 k  | frusta        | 2 min             | 45 min        | 86 min  | 30.15| 0.89 | 0.24  |
> > | 2 k    | ours (CNNO+QF)| 2.1 min           | 12 min        | 194 min | 26.6 | 0.85 | 0.16  |
> > | 2 k    | frusta        | 20 min            | >24 h         | >24 h   | -    | -    | -     |
> >
> > We implement the frusta algorithm with the following steps:
> > 1. The intrinsic matrix of camera is obtained to calculate the near-far plane.
> > 2. The frusta of cameras is transformed to world coordinate system.
> > 3. We check the intersection of their bounding box to judge the view intersection.
> >
> > **Analysis**:
> > 1. (Universality) Notably, our matching strategy only needs the position and orientation of coarse cameras, i.e., only the 3-rd, 4-nd colume of extrinsic matrix. Moreover, we do not need the information of intrinsic matrix. This indicates the universality of our method.
> > 2. (Efficiency) As shown in QF section and Appendix C, we calculate the relative orientation with only 1 inner product and 1 cross-product based on fully convincing mathematical proof. As shown in column of pairs computing time, this way greatly improves our efficiency compared to frusta method, which contains time-consuming coordinate transform and intersection checking process.
> > 3. (Reconstruction Quality) In terms of reconstruction quality, there is no obvious difference between our method and frusta, since pose accuracy is well-solved in BA procedure. Our approach offers better compatibility in cases where initial poses are poor.
> > 4. (Reason why frusta failed) Though the naive frusta method sounds reasonable, however, in practice, especially in the driving scenario with long and narrow camera track, the frusta method cannot filter image pairs efficiently since all of the camera orientation are extremely similar.
> >
> > > **Q5: Comparison provided with original 3DGS instead of other state-of-the-art methods.**
> >
> > **A:** As shown in the table below, our framework can seamlessly integrate other 3DGS improvement methods and significantly enhance performance. The experiments are conducted on the Waymo dataset. This primarily relies on our graph-guided optimization module.
> > |          Methods                  |    PSNR           |    SSIM           |   LPIPS           |
> > |:-------------------------        |:---------:        |:---------:        |:---------:        |
> > | Scaffold w/ Original Pose         |   23.67           |   0.784           |   0.320           |
> > | Scaffold w/ Colmap                |   32.02           |   0.921           |   0.206           |
> > | Scaffold w/ Ours                  |  33.9         | 0.923         | 0.205         |
> > | 2DGS w/ Original Pose             |   19.39           |   0.624           |   0.705           |
> > | 2DGS w/ Colmap                    |   28.85           |   0.879           |   0.284           |
> > | 2DGS w/ Ours                      | 31.88         | 0.902         | 0.236         |
> > | OctreeGS w/ Original Pose         |   21.83           |   0.726           |   0.404           |
> > | OctreeGS w/ Colmap                |   29.03           |   0.870           |   0.295           |
> > | OctreeGS w/ Ours                  | 31.90         | 0.903         | 0.252         |
> >
> > To clearly distinguish our contributions and ensure the fairness of comparisons, we opt not to publish results in paper based on other latest state-of-the-art (SOTA) 3DGS methods. Additionally, because we do not segment into chunks, we can not directly apply our method to Hierarchical 3D Gaussian representations. Relevant results will be added in a future version.

---

> ### Author Response · Authors · 2024-11-21
>
> > **Q6:  Provide results on the MipNeRF360 dataset.**
>
> **A:** We consider the comparison with the original 3DGS not as a direct comparison, but as a type of ablation study. Our framework can be seamlessly integrated into methods such as 3DGS, 2DGS, Scaffold, OctreeGS, etc. Due to time constraints, all other results come from those published in OctreeGS. We provide the results with the original 3DGS method on the Mip360 dataset as follows:
> |    Methods            |    PSNR           |    SSIM           |   LPIPS           |
> |:-------------:        |:---------:        |:---------:        |:---------:        |
> | 3D-GS                 | 27.54             | 0.815             | 0.216             |
> | Mip-NeRF360           | 27.69             | 0.792             | 0.237             |
> | 2D-GS                 | 26.93             | 0.800             | 0.251             |
> | Mip-Splatting         | 27.61             | 0.816             | **0.215**             |
> | Scaffold-GS           | 27.90             | 0.815             | 0.220             |
> | Octree-GS             | 28.05             | 0.819             | 0.214             |
> | Ours w/ 3DGS          | **28.71**         | **0.868**         | 0.216         |
>
> > **Q7: Contributions are dispersed and not fully evaluated with SoTA methods.**
>
> **A:** Thank you for your positive feedback on the improvements highlighted in our paper. Please be clarified that our research is dedicated to reconstructing outdoor 3DGS scenes from unaligned images, targeting the complexities of 3DGS and outdoor settings. As shown in Q5, our graph-guided optimization module has a significant improvement over various SOTA methods. To address your concerns, we conducted additional experiments and will include these comparisons in future versions.
>
> > **Q8: What does "w/o structure estimation" mean in Table 4.**
>
> **A:** This means we only use Waymo’s original poses, not the estimated pose by our structure estimation module, for 3DGS reconstruction.
>
> > **Q9: Which datasets are used for results in Table 4, Table 5, and Table 7.**
>
> **A:** Tables 4, 5, and 7 all use the Waymo datasets for evaluation, with each segment containing 600 images.
>
> > **Q10: Dust3r Details.**
>
> **A:** We solely use Dust3R’s pairwise pose estimation.
>
> > **Q11: Definition of S_3^i in Eqn 1.**
>
> **A:** In the CNNP part, we traverse all of the cameras (from 1 to N, supposing current camera as c_i) and calculate which camera (c_j) should form a matching pair with c_i. S_3 means for c_i, we always add c_(i-1) in the traversal loop to form a matching pair. This can guarantee the camera graph must be a connected graph as illustrated in Remark.
>
> > **Q12: Definition of a Large-Scale Scene.**
>
> **A:** To avoid confusion, we will remove it.
>
> > **Q13: Clarification of Formula 8.**
>
> **A:** Thank you for pointing that out; we will make the corrections.
>
>
>
> **Reference**
>
> [R1] J. Wang et al., “Visual Geometry Grounded Deep Structure from Motion,” arXiv preprint arXiv:2312.04563, 2023. [Online]. Available: https://arxiv.org/abs/2312.04563
>
> [R2] L. Pan et al., “Global Structure-from-Motion Revisited,” arXiv preprint arXiv:2407.20219, 2024. [Online]. Available: https://arxiv.org/abs/2407.20219
>
> [R3] E. Brachmann et al., “Scene Coordinate Reconstruction: Posing of Image Collections via Incremental Learning of a Relocalizer,” arXiv preprint arXiv:2404.14351, 2024. [Online]. Available: https://arxiv.org/abs/2404.14351
>
> [R4] P. Lindenberger et al., “Pixel-Perfect Structure-from-Motion with Featuremetric Refinement,” arXiv preprint arXiv:2108.08291, 2021. [Online]. Available: https://arxiv.org/abs/2108.08291

---

### Author Response · Authors · 2024-11-25

Dear Reviewers

Thank you for your efforts in reviewing our submission. As the rebuttal deadline approaches, your feedback would be greatly appreciated and highly valuable. We have carefully addressed the comments received so far and hope for the opportunity to engage with your insights as well. Your comments are invaluable for improving our work and fostering meaningful discussions.

Thank you again for your time and contribution to the review process!

Best regards

---

### Comment · Area_Chair_aKaV · 2024-11-25
**Last day for interactive discussions!**

Dear authors and reviewers,

The interactive discussion phase will end in one day (November 26). Please read the authors' responses and the reviewers' feedback carefully and exchange your thoughts at your earliest convenience. This would be your last chance to be able to clarify any potential confusion.

Thank you,
ICLR 2025 AC

---

### Comment · Reviewer_4dAT · 2024-11-26
**Thoughts on new experiments and results**

I would like to thank the authors for answering many of the questions with concrete experimental results in such a short period of time. Based on the new results shared by the authors, many of my concerns are satisfied. It is particularly promising that the proposed collection of ideas improve GS quality in conjunction with multiple methods apart from vanilla 3DGS. I think the SfM runtime comparison with other baselines also provide good evidence that the improvements add to efficiency in a notable manner.

Given the new experiments and results, my main suggestion to the author would be to pay attention to the framing and premise of the proposed method. In my opinion, this is a paper of high practical value, it's a number of small algorithmic changes that collectively lead to notable improvements. I recommend authors to provide as much detail as possible of their practical setup, empirical evidence, honest discussions of any limitations they observed (for example authors shared the challenges brought on by differences between autonomous driving datasets vs. more typical 360 datasets), etc. Such discussion will make the paper stronger and add to its value.

---

> ### Author Response · Authors · 2024-11-26
>
> We thank you for your valuable comments and feedback. We are pleased to learn that you recognize the contributions of our proposed method in enhancing the quality and efficiency of GS, and we appreciate your affirmation of our new experimental results.
>
> We greatly appreciate your valuable suggestion to focus on the framework and premises of our paper. We understand the importance of clearly presenting the motivation and overarching approach of our method to our readers. To this end, we will refine our paper in the following areas:
> > **Q1: Detailed description of the experimental setup.**
>
> We will incorporate the following experimental details into our paper.
> Experiments were conducted using an NVIDIA RTX 3090 GPU and an AMD EPYC 7542 CPU. Dust3R was utilized for initial pose estimation across both the Waymo, Kitti, and Mill 19 datasets, achieving pose estimation within 0.01 seconds per pair. The Waymo includes 600 images from three viewpoints (left, center, right), covering 599 image pairs. The Kitti includes 100 images with 99 image pairs. The Mill 19 comprises 2000 images with 1999 pairs included.
>
> For the CNNP configuration, settings of “r=5, h=20, w=1” were employed. “r=5” designates the five nearest cameras as matching candidates for each camera  c_i . The “h=20, w=1” configuration means that one camera is chosen as a match for  c_i  out of every 20 cameras based on proximity. To optimize multi-view consistency and balance the coefficients, we set the coefficient \lambda to 0.07, which was empirically found to be optimal in our experiments. Additionally, we adopted a graph-guided optimization setup, where the sampling probability is determined by the weights of the graph nodes. We set the minimum sampling probability at 0.5 to ensure that nodes with lower weights are not overlooked.
>
> For comparative experiments with Colmap, the Colmap setup included a vocabulary tree containing 256K visual words, pre-built using the Flickr100k dataset (available on the COLMAP project page).
>
> > **Q2: Enhanced Presentation of Empirical Evidence.**
>
> We have added more experimental results, including comparisons with other SFM baseline methods and the outcomes of applying our method to various 3DGS methods. These data further confirm the effectiveness and practicality of our approach, which will be included in a future version of the paper to enhance its persuasiveness.
>
> > **Q3: Discussion of limitations and challenges.**
>
> As you suggested, we will add a section discussing the limitations and challenges of our method.
>
> Our research focuses on 3DGS reconstruction of outdoor, unbounded scenes, primarily facing two challenges: 1) Accurate pose estimation outdoors is difficult due to the unpredictability and complexity of outdoor environments; 2）Outdoor scenes generally have sparser camera coverage and less overlap between images compared to indoor settings, resulting in insufficient constraints during training. To address these challenges, we introduced two key modules: Spatial Prior-Based Structure Estimation and a graph-guided Gaussian optimization strategy. These modules are designed to efficiently and accurately complete the reconstruction of outdoor scenes.
>
> However, our method shows limited improvement in 3DGS reconstruction of objects. This is primarily because datasets in this category usually have precise ground truth (GT) poses, and the camera setups often rotate 360 degrees around the object, providing sufficient image overlap and constraints to aid convergence. This results in nearly equal importance weights for graph nodes, which diminishes the impact of our optimization.
>
> Moreover, our approach relies on the accuracy of rough spatial priors. If the initial camera pose distribution in the xy plane is inaccurate, it may lead to Bundle Adjustment (BA) failure. In such cases, our method might not effectively handle pose estimation errors.
>
>
>
>
> **We value the insights your feedback has provided and are committed to continually refining our work. By conducting additional experiments and evaluations, we believe we can demonstrate that our contributions are substantial and highly relevant to practical 3DGS reconstruction tasks. We look forward to further discussions and appreciate the opportunity to strengthen our submission.**

---

### Comment · Reviewer_MKvT · 2024-11-26

Thank the authors for their detailed rebuttal. Based on the responses, most of my concerns have been addressed. This work focuses more on large-scale pipeline improvements, introducing a series of practical enhancements, particularly in the SFM component to provide better initial values for GS with faster speed. While its technical novelty is relatively limited, it offers valuable practical contributions.

---

> ### Author Response · Authors · 2024-11-27
>
> Dear Reviewer,
>
> Thank you for providing valuable feedback. We are pleased to know that our responses have addressed most of your concerns and appreciate your recognition of our pipeline improvements.
>
> We would like to take this opportunity to further clarify and emphasize the technical contributions and novelty of our work:
> 1. We propose a novel spatial prior-based structure estimation method, where the proposed Quadrant Filter rapidly computes the 6-bit encoded relative position and orientation. It significantly improves estimation speed and robustness in outdoor settings.
>
> 2. We are the first to involve graph information into the 3DGS optimization process, such that the overfitting caused by limited constraints in outdoor scenes is mitigated. This strategy significantly enhances reconstruction quality and is applicable to various 3DGS-based methods.
>
> We present extensive experiments and convincing mathematical proof. Our method can also effectively process uncalibrated images from the internet to reconstruct outdoor scenes.
>
> Thank you for your attention and feedback. We hope this clarifies concerns regarding our method’s novelty and highlights our contributions to the field. We appreciate your reconsideration of our work based on these points.

---

### Comment · Reviewer_X4qU · 2024-11-27

I would like to thank the authors for addressing all review comments in detail and providing additional evidence and clarifications about their method. Please consider revising your camera-ready version based on the review comments to enhance the readability and focus of the paper.

---

> ### Author Response · Authors · 2024-11-27
>
> Thank you for your positive feedback and constructive comments. We appreciate your recognition of our clarifications and proposed ideas.
> In response to your suggestions, we will enhance the clarity and precision of our method descriptions, provide detailed methodological and experimental information for reproducibility, and reinforce the presentation of our results to emphasize the efficacy and practical utility of our approach.
> We value your guidance and are committed to presenting an improved version.

---

### Meta-Review · Area_Chair_aKaV · 2024-12-19

**Metareview:**

The submission received positive reviews from all the reviewers. The reviewers generally appreciate the clarity and recognize the significance and experiments of the work. After reading the paper, the reviewers' comments and the authors' rebuttal, the AC agrees with the decision by the reviewers and recommends acceptance.

**Additional Comments On Reviewer Discussion:**

The reviewers raised questions regarding insufficient experimental comparisons (4dAT, MKvT, X4qU) and clarity on details (4dAT, MKvT, X4qU). The questions were adequately addressed by the authors. The AC agrees with the reviewers' evaluation that the paper should be accepted.

---

### Decision · Program_Chairs · 2025-01-22

Accept (Poster)